# Multidrug Resistance Like Protein 1 Activity in Malpighian Tubules Regulates Lipid Homeostasis in *Drosophila*

**DOI:** 10.3390/membranes11060432

**Published:** 2021-06-08

**Authors:** Wen Liu, Hao Cao, Moses Kimari, Georgios Maronitis, Michael J. Williams, Helgi B Schiöth

**Affiliations:** 1Functional Pharmacology, Department of Neuroscience, Uppsala University, 751 24 Uppsala, Sweden; caohao0311@163.com (H.C.); moses.kimari@neuro.uu.se (M.K.); Georgios.Karamanis@neuro.uu.se (G.M.); michael.williams@neuro.uu.se (M.J.W.); 2Institute for Translational Medicine and Biotechnology, Sechenov First Moscow State Medical University, 119991 Moscow, Russia

**Keywords:** lipid metabolism, ABCC1, kidney, oxidative stress, xenobiotic

## Abstract

**Simple Summary:**

Multidrug resistance proteins (MRPs) are important for ion transport, toxin/xenobiotic secretion, and signal transduction. Although studies have been undertaken to understand their physiological function, it is not fully known how MRPs may regulate metabolism. We knocked down the expression of *Drosophila multidrug-resistance like protein 1* (*MRP*) in several tissues central to metabolic regulation. Reducing MRP in Malpighian tubules, the functional equivalent to the human kidney, was sufficient to disrupt metabolic homeostasis, owing to abnormal lipid accumulation, as well as changes in feeding behavior. It also increased oxidative stress resistance in adult flies, possibly due to reduced levels of reactive oxygen species.

**Abstract:**

Multidrug resistance proteins (MRPs), members of the ATP-binding cassette transporter (ABC transporter) family, are pivotal for transporting endo- and xenobiotics, which confer resistance to anticancer agents and contribute to the clearance of oxidative products. However, their function in many biological processes is still unclear. We investigated the role of an evolutionarily conserved MRP in metabolic homeostasis by knocking down the expression of *Drosophila multidrug-resistance like protein 1* (*MRP*) in several tissues involved in regulating metabolism, including the gut, fat body, and Malpighian tubules. Interestingly, only suppression of *MRP* in the Malpighian tubules, the functional equivalent to the human kidney, was sufficient to cause abnormal lipid accumulation and disrupt feeding behavior. Furthermore, reduced Malpighian tubule *MRP* expression resulted in increased *Hr96* (homolog of human *pregnane X receptor*) expression. *Hr96* is known to play a role in detoxification and lipid metabolism processes. Reduced expression of *MRP* in the Malpighian tubules also conveyed resistance to oxidative stress, as well as reduced normal levels of reactive oxygen species in adult flies. This study reveals that an evolutionarily conserved MRP is required in *Drosophila* Malpighian tubules for proper metabolic homeostasis.

## 1. Introduction

In many species, the ATP-binding cassette transporters (ABC transporters) are one of the largest transporter families. These transporters take part in cellular substrate influx and efflux. Among ABC transporters, the multidrug resistance proteins (MRPs), also known as ABCCs and GS-X pumps, belong to the C subfamily and have been shown to contribute to ion transport, toxin/xenobiotic secretion, and signal transduction [1,2]. Due to the ubiquity of *MRP* expression, numerous studies have been performed on their physiological functions [3,4].

The evolutionarily conserved MRP family member *multidrug resistance protein 1* (*MRP1*) is widely expressed in metabolic and excretory tissues, including lung, kidney, intestines, and adipose [5,6]. In vertebrates, *MRP1* is well known for participating in the transport of steroid hormones, lipophilic anions, glutathione conjugates, and various drugs and xenobiotics [5,6,7,8]. However, *MRP1* knockout mice are viable and fertile, indicating that *MRP1* is not required for embryonic developmental processes [9]. On the other hand, abnormal *MRP1* expression is remarkably relevant for various diseases and biological processes, such as hemolysis and various cancers [10,11,12].

The human kidney, which is the central metabolite excretion and reabsorption tissue, expresses *MRP1* at high levels, and changes in *MRP1* expression within the renal system are associated with chronic renal failure, inflammation, and problems with water reabsorption [13]. Moreover, human MRP1 has been reported to affect a wide range of physiological processes, including drug/xenobiotic clearance, oxidative stress, and inflammatory responses [14,15,16].

Recently, MRPs were shown to be involved in metabolic disruptions and syndromes in adult mice, including nonalcoholic fatty liver disease and obesity [17,18]. Furthermore, it was reported that MRP1 protein levels are significantly increased in the glomeruli of diabetic rats [19]. Therefore, it is reasonable to speculate that MRP proteins are associated with not only substrate transport but also other organic processes. Moreover, in a previous study, we reported that feeding *Drosophila* the xenobiotic dibutyl phthalate throughout larval development induces abnormal lipid accumulation, as well as causing a reduction in *MRP* transcript levels [20]. As a crucial regulator of xenobiotic transport, *MRP* may also contribute to disruptions in metabolic homeostasis.

Therefore, in order to better understand the function of MRP1 in the regulation of metabolic homeostasis, we employed the genetically tractable organism *Drosophila melanogaster* (the common fruit fly). *Drosophila multidrug-resistance like protein 1* (*MRP*) is highly homologous and functionally similar to human MRPs 1, 2, and 3. In *Drosophila*, the Malpighian tubules are the main excretory organ and are the physiological equivalent of the vertebrate kidney. Flies have four Malpighian tubules that connect at the conjunction of the midgut and hindgut, and the interaction between the intestine and the Malpighian tubules is necessary for metabolite elimination and homeostasis [21]. Similar to vertebrate kidneys, *Drosophila MRP* is highly expressed in the Malpighian tubules [22]. In this study, we investigated the tissue-specific effects of MRP on *Drosophila* metabolism.

## 2. Material and Methods

### 2.1. Fly Strains and Maintenance

The w*; P{w[+mC]=Uro-GAL4.T}2, (Uro-GAL4); w*; P{w[+mW.hs]=GawB}48Y, (48Y-GAL4); w*; P{w[+mW.hs]=GawB}c601[c601], (c601-GAL4); w^*^; P{w[+mC]=ppl-GAL4.P}2, (ppl-GAL4), and y^1^ sc* v^1^; P{TRiP.HMS01780}attP2, (UAS-MRP RNAi) were provided by the Bloomington Stock Center (Bloomington, IN, USA). All fly stocks and crosses were maintained on Jazz-mix Drosophila food (Fisher Scientific, Gothenburg, Sweden) and supplemented with yeast extract (Genesee Scientific, San Diego, CA, USA). These conditions will not affect the flies’ lifespan. The flies were raised at 25 °C and 60% humidity on a 12:12 h light/dark cycle. The F_1_ progeny were collected immediately after eclosion and aged for 5–7 days before all assays. Due to the fact the frequent reproductive cycle of the female flies could potentially influence their behavior, we used only males in our experiments.

### 2.2. Knockdown of MRP

MRP was knocked down in a transgenic cross and checked by performing quantitative RT-PCR (qPCR). Expression of the RNAi construct for MRP was driven by the GAL4-UAS system using the neuronal-specific elav-GAL4 driver. Thus, RNAi to MRP was expressed in all neurons. Whole male flies aged 5–7 days posteclosion were collected for analysis. The experiment was repeated five times and 10 male flies were used for each replicate.

### 2.3. Starvation Assay

The *Drosophila* Activity Monitoring System (DAMS) from TriKinetics (Waltham, MA, USA) was employed to analyze the starvation resistance. Flies were collected after eclosion and aged in a vial with 8 mL of food. After 5–7 days of aging, each individual fly was placed in a 5 mm diameter tube and then monitored using the DAMS. The tubes were filled with 1% agarose (~2–3 cm height), which provided water and humidity but not an energy source for the flies during the assay. Then a black cap was placed on the agarose-containing end. Flies were transferred to the open side, and after they were placed in the tubes, the opening was blocked with a cotton plug. Each genotype was monitored by one DAMS, and each DAMS held 32 flies. As described, the alive/dead status of the flies was measured by the beam-crossing numbers [23]. The starvation resistance was calculated as the average survival time under starvation.

### 2.4. Carbohydrate Assay

Concentrations of body trehalose and glycogen were assessed using the Liquick Cor-Glucose diagnostic kit (Cormay, Marynin, Poland). At least 6 biological replicates from both control groups and the experimental group were prepared. For each replicate, 10 mg of male flies were decapitated and homogenized in 100 μL PBS buffer (pH 7.4). The samples were deproteinized at 70 °C for 5 min, followed by incubation on ice for 5 min. The homogenized pellets were then centrifuged at 12,900× *g* at 4 °C for 15 min. The supernatant was collected, and 10 μL of body supernatant was mixed with either 10 μL of 2 μL/mL porcine kidney trehalase (Sigma-Aldrich, Stockholm, Sweden) to digest trehalose or 10 μL of 1 mg/mL amyloglucosidase from *Aspergillus niger* (Sigma-Aldrich, Stockholm, Sweden) to digest glycogen to glucose. The trehalose samples were incubated at 37 °C, and the glycogen samples were incubated at 25 °C, overnight. After incubation, 10 μL of sample was diluted in 90 μL of deionized water and mixed with 650 μL of glucose reaction mixture (Cormay, Marynin, Poland) in a 96-well plate. Absorbance was measured with a Multiscan GO spectrophotometer (Thermo Scientific, Stockholm, Sweden) at 500 nm. For producing the standard curve, a serial dilution was made from a 10 μg/μL standard glucose solution. The concentration was calculated according to the produced standard curve.

### 2.5. Triacylglyceride (TAG) Assay

Five flies were collected for each biological sample, and 6–8 replicates were prepared for all control and experimental groups. The flies were homogenized in 100 μL of cold PBST (pH = 7.2, 0.05% of Tween 20). After 10 min incubation at 70 °C, 20 μL of the homogenized sample was separately added to either 20 μL of PBST, for measuring the free glycerol, or 20 μL of triglyceride reagent (Sigma-Aldrich, Stockholm, Sweden), for measuring the total glycerol content. All samples were then incubated at 37 °C for 60 min and then centrifuged for 3 min at 14,000 rpm. After incubation, 30 μL of each individual sample was transferred to a 96-well plate and mixed with 100 μL of free glycerol reagent (Sigma-Aldrich, Stockholm, Sweden). The plate was incubated for 5 min at 37 °C. The absorbance of each sample was measured with a Multiscan GO spectrophotometer (Thermo Scientific, Stockholm, Sweden) at 540 nm and calculated according to the standard curve. The standard curve was obtained from a serial dilution of glycerol standards (Sigma-Aldrich, Stockholm, Sweden) and was produced along with the samples. The TAG concentration was determined as *TAG concentration* = *total glycerol concentration* − *free glycerol concentration.*

### 2.6. Fly Proboscis and Activity Detector (FlyPAD)

We performed FlyPAD experiments using starved flies because the food pellet can only hold a small amount of food (150 mM sucrose, only 2–3 microliters), which will dry out within 2 h, resulting in feeding times of usually no more than 2 h. Fed flies usually eat slowly; therefore, we chose to use flies starved for 18 h instead. The assay was adapted from Itskov et al. [24]. In brief, male flies from both control and experimental crosses were collected after eclosion and aged for 5 days, under a 12:12 h light/dark cycle, at 25 °C. The assay was conducted between 8 a.m. and 11 a.m. Before the experiment was performed, flies were transferred into the flyPAD arena (flies were not anesthetized before transfer). Each arena included two detective channels, one with a 150 mM sucrose (dissolved in 1% agarose) pellet and one that was blank. All the interactions between the flies and the food pellet were monitored by Bonsai software. Thirty-two replicates were used for each group. Data were analyzed by using the MATLAB-based graphic user interface (GUI) program.

### 2.7. ROS Detection

Reactive oxygen species were monitored using the Amplex Red Hydrogen Peroxide/Peroxidase Assay Kit, according to the instruction provided by the manufacturer (Molecular Probes, Eugene, OR, USA). Six fly bodies were homogenized in phosphate buffer (pH 7.4) and used as one replicate. The reaction mixture contained 50 μM Amplex Red reagent and 0.1 U/mL horseradish peroxidase (HRP). Fluorescence was recorded at 530 nm excitation/587 nm emission. The results were normalized using *Uro-GAL4* > *w*^1118^ control flies.

### 2.8. Paraquat Resistance

Flies were collected and raised in a vial containing 8 mL of normal fly food at 25 °C for 5 posteclosion days and then transferred to a vial containing 1% agarose for 6 h starvation in order to eliminate all the remaining food. Paraquat (Sigma-Aldrich, Stockholm, Sweden) was dissolved in 10% (*w*/*v*) sucrose solution at a final concentration of 20 mM. A round filter paper soaked with paraquat solution was placed in a new agarose vial, and 13–15 flies for each replicate were transferred to this vial after starvation; 6–8 biological replicates were prepared. To assess the resistance to paraquat, the survival rate was recorded 3 times per day until all flies died, and both the filter papers and the agarose vials were replaced daily.

### 2.9. Quantitative Real-Time PCR

Six replicates containing 10 decapitated fly bodies for each group were prepared. The RNA was extracted by using the TRIzol reagent (Invitrogen, Stockholm, Sweden). cDNA reverse transcription was performed by using a High-Capacity RNA-to-cDNA kit (Applied Biosystems, Foster City, CA, USA). Approximate 10 ng of cDNA sample was used as a template for each replicate and amplified with Taq DNA polymerase (Biotools) in Bio-IQ5 cycler (Bio-Rad, Solna, Sweden). Rp49 was used as the reference gene. The primer sequences used were as follows:Rp49-F: 5′-CACACCAAATCTTACAAAATGTGTGA-3′;Rp49-R: 5′-AATCCGGCCTTGCACATG-3′;Hr96-F: 5′-GATATGTTCCTCCAGGCCCTA-3′;Hr96-R: 5′-TGTGCGTGGCAAAGAAGACT-3′;Cnc-F: 5′-CTGCATCGTCATGTCTTCCAGT-3′;Cnc-R: 5′-AGCAAGTAGACGGAGCCAT-3′;Keap1-F: 5′-AGGCCAATGTGTTTATTGAGCG-3′;Keap1-R: 5′-GCAATCAACTGATATGCCGAAAG-3′;ss-F: 5′-GATATGTTCCTCCAGGCCCTA-3′;ss-R: 5′-TGTGCGTGGCAAAGAAGACT-3′;MRP-R: 5′-GAATCTGGGTCTGCTGGTAATC;MRP-R: 5′-AAACATCCAGGTCGTAGAGCG.

## 3. Data Analysis

Data analysis and plotting were performed using Prism GraphPad 5. Data are presented as mean ± SEM. Except for the paraquat survival test, all the significant differences were tested following one-way ANOVA with Tukey’s post hoc test. For evaluating the paraquat resistance, the comparisons were analyzed by Kaplan–Meier log-rank test. The significances are represented as * *p* < 0.05, ** *p* < 0.01, and *** *p* < 0.001 and are also specified in the corresponding figure legends.

## 4. Results

### 4.1. Successful Knockdown of MRP

In order to determine if the RNAi line was functional, we crossed the UAS-MRP RNAi line with elav-GAL4 flies, to express the RNAi line on all neurons, and performed quantitative RT-PCR (qPCR) on the adult F_1_ progeny. Appendix A shows that MRP was successfully knocked down in the transgenic cross.

### 4.2. Loss of MRP in Malpighian Tubules Increases Starvation Resistance

*Drosophila MRP* is highly expressed in the gut and Malpighian tubules and at lower levels in the fat body [22,25]. To study the function of *MRP* in different organs, we crossed *UAS-MRP* RNAi flies with four tissue-specific GAL4 drivers, namely a mid-gut driver (*48Y-GAL4*), a hind-gut driver (*c601-GAL4*), a fat body driver (*ppl-GAL4*)*,* and a Malpighian tubule driver (*Uro-GAL4*), and then performed a starvation resistance assay using 5–7-day-old male flies from the F_1_ generation. It has been reported that *MRP* is expressed in all of these tissues [25]. The starvation resistance assay can potentially uncover any effects on energy utilization, lipid storage, and feeding behavior. Notably, we observed that none of the *48Y-GAL4*, *c601-GAL4,* and *ppl-GAL4* crosses exhibited a starvation resistance phenotype when compared to control flies (Figure 1A–C). However, knocking down *MRP* in the Malpighian tubules (*Uro-GAL4* > *UAS-MRP* RNAi) significantly increased resistance to starvation when compared to control flies (61.57 ± 1.16 h for *Uro-GAL4* > *UAS-MRP* RNAi vs. 56.66 ± 0.85 h for *Uro-GAL4* > *w*^1118^ (*p* < 0.05) and vs. 54.90 ± 1.47 h for *w*^1118^ > *UAS-MRP* RNAi (*p* < 0.001); Figure 1D).

### 4.3. Loss of MRP in Malpighian Tubules Influences Triacylglyceride Levels

The starvation resistance defect, found in *Uro-GAL4* > *UAS-MRP* RNAi flies, established a possibility for metabolic disruption. Therefore, we measured the carbohydrate and triacylglyceride (TAG) levels of adult flies. The levels of circulating glucose (Figure 2A), circulating trehalose (Figure 2B), stored trehalose (Figure 2C), or glycogen (Figure 2D), were not significantly affected by the reduction in *MRP* expression in Malpighian tubules. However, the TAG content of the experimental group (2.44 ± 0.29 mg/mL, *p* < 0.05, Figure 2E) was significantly elevated compared to control groups (1.23 ± 0.08 mg/mL for *Uro-GAL4* > *w*^1118^ and 0.57 ± 0.08 mg/mL for *w*^1118^ > *UAS-MRP* RNAi).

Sieber et al. reported that *Hr96*, the *Drosophila* homolog of human nuclear receptor subfamily 1, group I, member 2 (NR1I2, also known as *pregnane X receptor* (*PXR*)), was involved in lipid, but not carbohydrate, homeostasis in fruit flies [26]. Furthermore, as a phase II enzyme in xenobiotic transportation, NR1I2 has been shown to interact with the ABC transporter family [27,28]. Moreover, we have shown that similar to *MRP*, exposure of larva to the xenobiotic dibutyl phthalate significantly reduced *Hr96* transcript levels [20]. Therefore, using qPCR, we examined *Hr96* expression in whole adult flies where *MRP* was knocked down in the Malpighian tubules. Compared to controls, *Hr96* expression increased significantly when *MRP* was knocked down in the Malpighian tubules (approximately 1.1-fold increase compared to *w*^1118^ > *UAS-MRP* RNAi group (*p* < 0.001) and 0.5-fold increase compared to *Uro-GAL4* > *w*^1118^ group (*p* < 0.05)) (Figure 2F).

### 4.4. Loss of MRP in the Malpighian Tubules Influences Feeding Behavior

Since we detected deficiencies in starvation resistance and lipid metabolism, we determined if *MRP* knockdown in Malpighian tubules might also influence feeding behavior. To do this, we employed the Fly Proboscis and Activity Detector (flyPAD). The flyPAD enables real-time recording of food intake and feeding behavior with a high temporal resolution [24]. As a consequence of *MRP* knockdown in Malpighian tubules, the *Uro-GAL4* > *UAS-MRP* RNAi flies exhibited an obvious reduction in total food intake, which was represented by the total number of sips (bites) (226.9 ± 12.5 sips for *Uro-GAL4* > *w*^1118^ and 224.2 ± 28.4 sips for *w*^1118^
*> UAS-MRP* RNAi vs. 112.5 ± 12.4 sips for *Uro-GAL4* > *UAS-MRP* RNAi, *p* < 0.005) (Figure 3A). Additionally, their activity bouts, which reflect all interactions of the flies with the food pellet, and feeding bursts, which denote the number of meals flies have during the monitoring, were decreased. This indicated that experimental flies had significantly fewer meals and interactions with the food (for the comparison of all the groups, *p* < 0.01) (Figure 3B,C). Of note, although the number of sips per feeding burst (Figure 3D) and the duration of each activity bout (Figure 3E) were unaffected, the meal duration (feeding burst) increased in the experimental flies (for the comparison of all the groups, *p* > 0.05) (Figure 3F).

### 4.5. MRP Knockdown in Malpighian Tubules Confers Oxidative Resistance to Drosophila melanogaster

MRP transporters are tightly related to oxidative responses [8,29], and a reduction in oxidative resistance may contribute to imbalanced metabolic homeostasis. Therefore, we performed an oxidative resistance assay by exposing flies to paraquat-containing food. Paraquat is a commonly used herbicide that can induce oxidative stress when administrated to flies [30]. The surviving flies were counted, and the survival rate was calculated daily; these data reflect the degree to which the flies were resistant to oxidative stress. To our surprise, the oxidative resistance of MRP knockdowns was not diminished, as we had assumed. In contrast, the *Uro-GAL4* > *UAS-MRP* RNAi flies showed increased resistance to oxidative stress. Unlike a report that showed the detectable susceptibility to oxidants of global *Drosophila MRP4* knockdowns [31], the Malpighian tubule specific *MRP* knockdown flies were more resistant to oxidative stress. The experimental flies survived significantly longer than both control groups (the median survival was 110 h for *Uro-GAL4* > *w*^1118^ and 68 h for *w*^1118^
*> UAS-MRP* RNAi, while the value was 136 h for *Uro-GAL4* > *UAS-MRP* RNAi, and the *p*-values were less than 0.01 and 0.001, respectively) (Figure 4A). In order to confirm this possibility, we conducted a reactive oxygen species (ROS) detection assay using freshly made whole-body homogenate. Consistent with the paraquat assay, the *Uro-GAL4* > *UAS-MRP* RNAi flies exhibited less ROS production than controls (Figure 4B).

Next, we used qPCR to examine the expression levels of genes associated with oxidative stress, including *ss* (*spineless*, the *Drosophila* homolog of *aryl hydrocarbon receptor* (*AhR*)) [32], *cnc* (*cap-n-collar*, the Drosophila homolog of *NF-E2-related factor 2* (*NRF2*)), and *Keap1* (*Kelch-like ECH associated protein 1*) [33,34]. The expression of *ss*, but not *cnc* or *Keap1*, was significantly increased in the *MRP* Malpighian tubule knockdowns (1.56 ± 0.16-fold change for *Uro-GAL4* > *UAS-MRP* RNAi, compared to 1.02 ± 0.07-fold change for *Uro-GAL4* > *w*^1118^ and 0.80 ± 0.05-fold change for *w*^1118^ > *UAS-MRP* RNAi) (Figure 4C).

## 5. Discussion

We determined that a specific reduction in *MRP* expression within the *Drosophila* Malpighian tubules is sufficient to increase triacylglycerol (TAG) levels in adult flies. Reduced *MRP* expression in the Malpighian tubules also disrupts normal feeding behavior and increases resistance to oxidative stress. This study highlights that ABC transporters, which are generally reported to be more involved in drug, xenobiotic, and ion transport, may also be important for regulating biochemical metabolites, such as lipids [35].

We also provide some evidence that *MRP* is an indispensable molecule in biochemical metabolism in *Drosophila*. To study the tissue-specific functions of *MRP*, we employed the efficient GAL4-UAS system [36] in order to knock down *MRP* in the mid-gut, hind-gut, fat body, or Malpighian tubules. Interestingly, we found that when *MRP* was knocked down in the Malpighian tubules, which are functionally similar to the mammalian kidney, flies were significantly more resistant to starvation. This indicates the possible importance of *MRP* in metabolic homeostasis. In *Drosophila* a relationship between the Malpighian tubules and metabolic signaling has been reported. Söderberg et al. reported that insulin signaling within the Malpighian tubules was important for responses to metabolic stress [37]. Moreover, in the same article, knocking down *tachykinin-like receptor at 99D* (*TkR99D*) in Malpighian tubules led to increased resistance to both desiccation and starvation. In another study, MRP was shown to participate in tachykinin-evoked ATP release [38]. Taken together, it is reasonable to speculate that in the Malpighian tubules, MRP might contribute to resisting metabolic stress by regulating energy balance and metabolic homeostasis.

Encouraged by our starvation findings, we measured the carbohydrate and lipid contents of flies in which *MRP* expression was reduced in the Malpighian tubules. As a consequence of *MRP* reduction, TAG levels were significantly higher than in the control flies. In fact, MRP-mediated lipid transport has been reported in mammals. Raggers et al. reported that *MRP1* transfected pig kidney had increased short-chain sphingolipid outward transport [39]. The evidence of lipid transport mediated by MRP was also confirmed by Kamp and Haest, who reported that the presence of MRP could be implicated in phospholipid outward transport [40]. Furthermore, MRP family members have been shown to transport cholesterol-derived corticosteroids [6]. Therefore, it is possible that *Drosophila* MRP mediates lipid homeostasis.

The human *PXR*/*CAR* homolog in insects, *Hr96*, has been reported [41] to control lipid homeostasis in *Drosophila* [26,42]. As a xenobiotic-sensing nuclear receptor, *NR1I2*/*PXR* activation was able to alter energy utilization and lipid allocation, which eventually leads to metabolic disorders [43,44]. Recent studies have demonstrated that *NR1I2*/*PXR* regulates the expression of key proteins involved in endobiotic responses, such as the metabolic homeostasis of lipids, glucose, and bile acid (OLADIMEJI et al., 2018). Furthermore, a connection between *NR1I2*/*PXR* and *MRP* was found in human hepatocytes [45], and *MRP* was shown to be regulated by *PXR/CAR* (Wei et al., 2000; Maglich et al., 2002). Here, we provide another clue regarding an interaction between *Hr96* and *MRP*, where the expression of *Hr96* was affected by the reduction in *MRP* specifically in the Malpighian tubules. Subsequently, we found that feeding behavior and food intake are influenced by Malpighian tubule *MRP*, which could be a result of MRP-induced lipid accumulation disrupting energy balance and inhibiting food intake and feeding patterns.

We also observed that oxidative responses were strengthened by *MRP* deficiency in the Malpighian tubules (see Figure 4). Considering that glutathione and its conjugates, some of the main substrates of MRPs, are crucial in oxidative responses, we chose to evaluate the role of Malpighian tubule expressed MRP in oxidative stress. Two investigations carried out by Takahashi et al. indicated that MRP is important in protecting endothelial cells against oxidative stress [46,47]. Furthermore, another *Drosophila* study validated that an *MRP4* mutation could suppress general oxidative resistance and contribute to lifespan elongation [31]. In contrast, a recent study illustrated that *MRP1^-/-^* mutant mice had reduced ROS production in a streptozotocin-induced diabetic model and suggested *MRP1* as a novel target for diabetes treatment [10]. Different from the global mutation, our findings show that interfering with *MRP* expression in the Malpighian tubules is sufficient to confer oxidative stress resistance. We then tested genes known to be highly regulated by the oxidative response. Although the Keap1/Nrf2 signal mediates the cellular oxidative response and xenobiotic elimination [48,49], no change in the expression of these genes was detected in *MRP* Malpighian tubule knockdown flies. However, when *MRP* was knocked down specifically in Malpighian tubules, the expression of the human *AhR* homolog *spineless* (*ss*) was significantly upregulated. AHR is a transcription factor known to regulate the cytochrome P450 genes and participate in phase I xenobiotic clearance in mammals [50,51]. There are reports showing that mice treated with AHR inducers exhibited increased expression of *MRPs* in hepatic cells [52,53]. Thus, a defect in MRP function, a part of the phase III xenobiotic clearance system, might cause the compensatory overexpression of *ss* in flies, and the activation of *ss* (*AhR*) may be necessary to protect cells under oxidative stress. Therefore, knocking down *MRP* in Malpighian tubules could lead to compensatory *ss* expression, which then contributes to oxidative resistance in flies.

## 6. Conclusions

Our present study reveals a role for Malpighian tubule expressed *MRP* in metabolic regulation in *Drosophila*. The tissue-specific deficiency of *MRP* led to overall lipid accumulation and changes in the feeding pattern of *Drosophila* adult males. Furthermore, inadequate *MRP* expression results in an inhibition of oxidative stress and ROS production. Altogether, these findings provide a new insight to guide future studies of MRP in metabolic homeostasis.

## Figures and Tables

**Figure 1 membranes-11-00432-f001:**
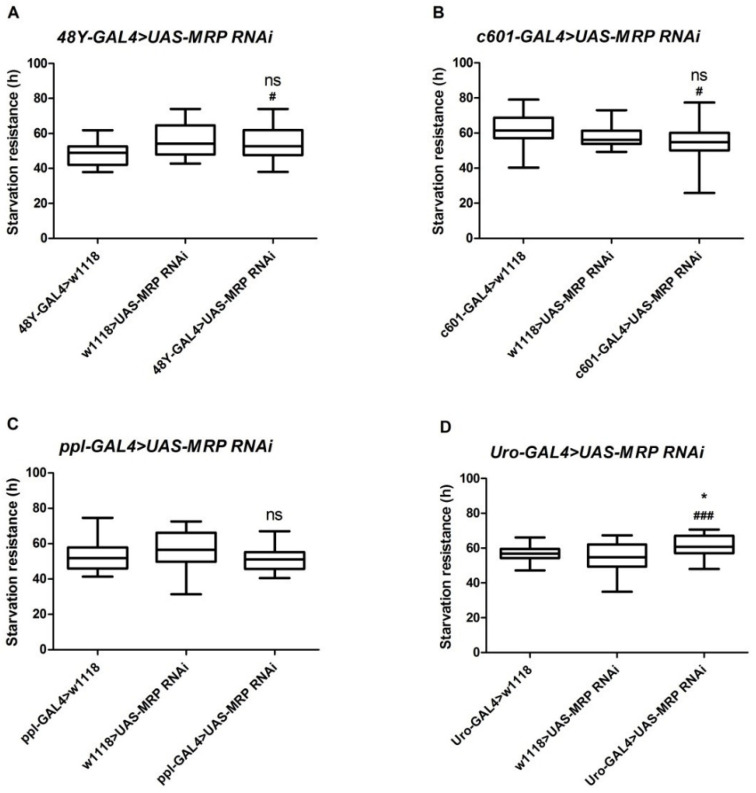
Crossing *MRP* RNAi with four different tissue-specific drivers. Graphs are presented in box plots, showing starvation resistance of both experimental and control groups. All Gal4 drivers were crossed with a *UAS-MRP* RNAi line: (**A**) *48Y-GAL4* (the mid-gut driver), (**B**) *c601-GAL4* (the hind-gut driver), (**C**) *ppl-GAL4* (the fat body driver), and (**D**) *Uro-GAL4* (the Malpighian tubule driver). Error bar represents the max and min values, and 32 male flies were used for each individual crossed strain; ns, not significant compared to either *w*^1118^ > *UAS-MRP* RNAi or both of the controls; ^#^
*p* < 0.01, compared to GAL4-driver > *w* ^1118^; ^###^
*p* < 0.001, compared to *Uro-GAL4* > *w*^1118^; * *p* < 0.01, compared to *w*^1118^ > *UAS-MRP* RNAi flies; one-way ANOVA with Tukey’s post hoc test was performed.

**Figure 2 membranes-11-00432-f002:**
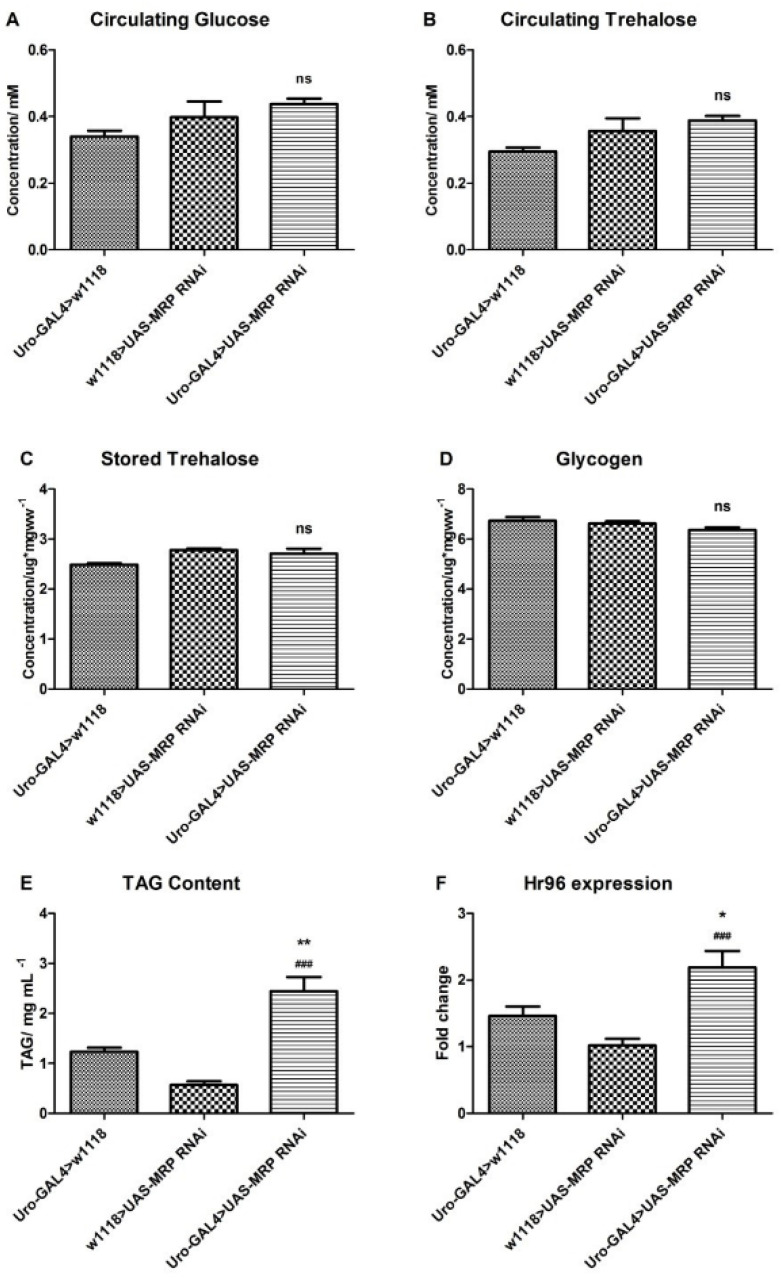
*MRP* knockdown in the Malpighian tubules affects lipid content. Flies for the carbohydrate assay were either fed *ad libitum* or starved for 24 h before the assay. *Uro-GAL4* > *UAS-MRP* RNAi flies exhibited no effects on their (**A**) circulating glucose, (**B**) circulating trehalose, (**C**) stored trehalose, and (**D**) glycogen. However, *MRP* knockdown in the Malpighian tubules induced a significant increase in (**E**) triacylglyceride (TAG) content. The mRNA expression of the xenobiotic sensing receptor, *Hr96*, was notably increased (**F**). All graphs are presented as mean ± SEM; 10–15 male flies for carbohydrate assay and 6 male flies for TAG assay were used per sample, and 4–6 biological samples were prepared for each assay; ns, not significant compared to either of the control groups; ** *p* < 0.01 and * *p* < 0.05, compared to *Uro-GAL4* > *w*^1118^; ^###^
*p* < 0.001, compared to *w*^1118^ > *UAS-MRP* RNAi flies; one-way ANOVA with Tukey’s post hoc test was performed.

**Figure 3 membranes-11-00432-f003:**
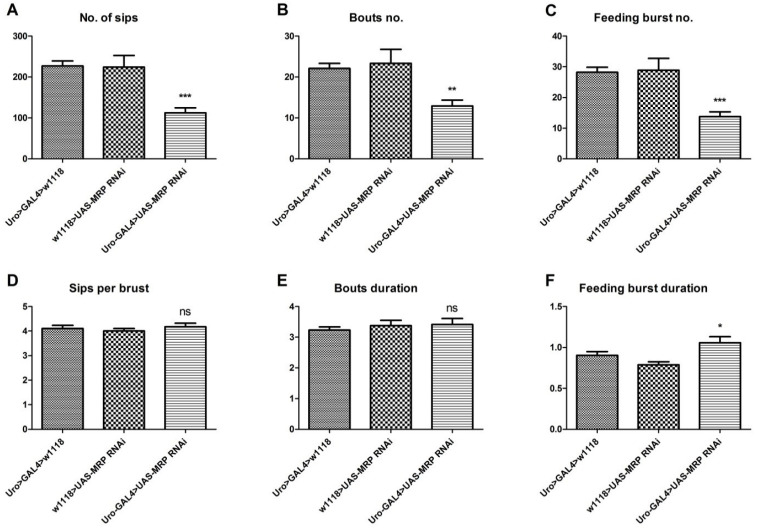
*MRP* knockdown in the Malpighian tubules affects feeding behavior. Male flies 5–7 days of age were used, and the flyPAD was employed to assess the feeding behavior. (**A**) The number of sips, which represents the total food intake; (**B**) the number of feeding bouts, which represents any interaction with food drop; and (**C**) the number of feeding bursts, which represents the number of meals, were all significantly decreased in the experimental group. However, both (**D**) the sips per burst and (**E**) bout duration were not affected. The feeding burst (**F**) duration was slightly increased in the experimental group. All graphs are presented as mean ± SEM, *n* = 32 males per group; ns, not significant; * *p* < 0.05, ** *p* < 0.01, *** *p* < 0.001; one-way ANOVA with Tukey’s post hoc test was performed; the experimental group was compared to each of the control groups.

**Figure 4 membranes-11-00432-f004:**
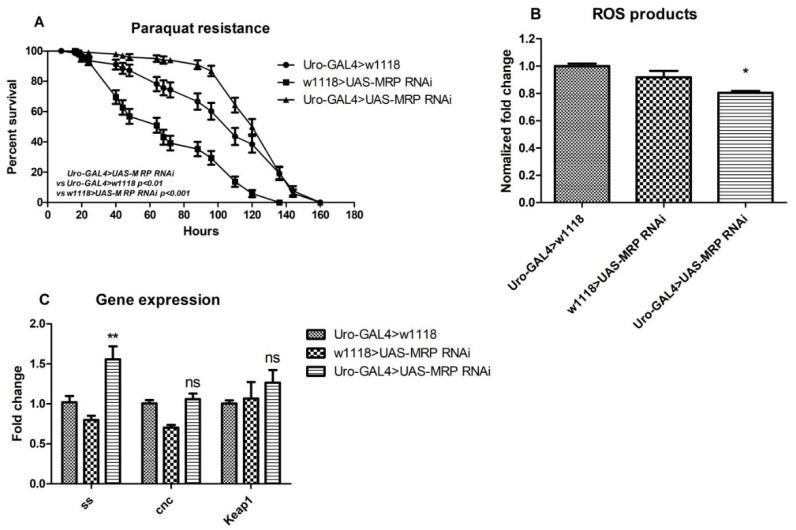
Malpighian tubule specific MRP knockdown conferred oxidative stress resistance. Flies 5–7 days of age were used. (**A**) Thirteen to fifteen flies were placed in a vial with a filter paper soaked with paraquat–sucrose solution. The death of flies was registered 3 times per day. The *Uro-GAL4* > *UAS-MRP* RNAi flies exhibited more resistance to paraquat than both of the control groups (6–8 replicates were used for each group of flies; data are represented as percentage survival ± SEM; *p* values were shown on the plot; Kaplan–Meier log-rank test was performed). (**B**) ROS production of the *Uro-GAL4* > *UAS-MRP* RNAi flies was significantly less than that of both of the controls, which indicates diminished oxidative stress level. Six flies were used for each replicate and 6 replicates were prepared for each group. The data were normalized to the *Uro-GAL4* > *w*^1118^ group. (**C**) The expression of *ss*, *cnc*, and *Keap1* were tested using qPCR. Although no difference was observed in *cnc* and *Keap1* expression, *ss* expression levels increased significantly in the experimental flies. The graph represents as mean ± SEM; * *p* < 0.05, ** *p* < 0.01; one-way ANOVA with Tukey’s post hoc test was performed.

## Data Availability

The authors confirm that the data supporting the findings of this study are available within the article.

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
