# Peer review of "Multidrug Resistance Like Protein 1 Activity in Malpighian Tubules Regulates Lipid Homeostasis in Drosophila"

_membranes, 2021, doi:10.3390/membranes11060432_

Round 1

Reviewer 1 Report

The authors have conducted a well-thought out study to explore the role of MRP-1 in Drosophila and its potential implications/value in humans. The way the study is designed, it demonstrates soundness of experimental design and they have tried to answer as many questions as possible. 

Some questions:

  1. What is the effect of MRP-1 knock-out on Drosophila? Does it affect the lifespan, male vs female differences, infection response? I would very much like it if the authors could add a sentence clarifying this.
  2. The authors have mentioned several times in the paper that lipid homeostasis is affected by MRP-1. As a visual demonstration (potentially in their next paper?), showing lipid droplets: size and distribution, by microscopy would be advantageous. 

Overall, this is a good study and I'm happy to endorse it for publication.

Author Response

Dear Reviewer,

We are very grateful for your positive opinion and valuable comments that helped to increase the quality of our work. Below in the attachment we present our responses to all comments, as well as the changes we introduced into the manuscript (marked with red). We truly believe these revisions have helped to make our manuscript higher quality. Thank you for your contribution.

Reviewer 2 Report

In this manuscript, Wen Liu et al. have studied the role of Multidrug Resistance like Protein 1 (MRP1) in Drosophila Malpighian tubules in lipid homeostasis. They have found that knocking down the expression of MRP1 in Malpighian tubules causes abnormal lipid accumulation and disrupts feeding behavior, which led to increased expression of detoxification-related protein and confers resistance to oxidative stress. The overall manuscript has been written and presented well. I have few comments below, and if authors can address those points, it will benefit readers and the scientific community.

Major points:

  1. Knocking down of MRP1 was done using Uro-Gal4, which surely labels Malpighian tubules cells, but at the same time, it also expresses in Gut and other cells in the fly body. Although Authors have used non-specific mid and hindgut drivers (48Y and c601-Gal4) to make sure that the effect of MRP1 reduction is not coming from the Gut, they have not used whole gut drivers like NP1-Gal4 (labeling all ECs) or DIMM-Gal4 (labeling all EEs). Against this backdrop, this remains inconclusive if their observed effect is coming from Malpighian tubules. It will be great if authors can provide an image of coverage and specificity of Uro-Gal4. This is the most straightforward experiment as they must combine Uro-Gal4 with UAS-GFP and image few organs like Malpighian Tubes, Gut and Brain.

  1. The second issue that authors have not mentioned in the manuscript is the developmental knockdown of MRP1 rather than knocking it down specifically in adults using the gal80ts system. The authors need to clarify why they utilized such a system, which may be confounded with compensatory effects during organism development. It will be best for their manuscript to perform at least one experiment with Uro-Gal4; Tub-gal80ts.

  1. The UAS-MRP-RNAi line has not previously been characterized for its efficiency; authors have assumed that it is working. Please use Q-PCR to verify knockdown efficiency. 

  1. TAG measurement is well presented; how about neutral lipids? Authors could have used Nile-red or Oil-red-O and image Malpighian tubules. This kind of experiment could have enhanced their paper quality and appeal.

  1. FlyPad data is good. However, it’s not clear from their data if they performed experiments in starved or fed flies. Presented data appears to be from hungry flies.

  1. Also, the authors show that MRP1 reduced flies are starvation resistant; from where the feeding effect is coming?

  1. Not really clear why the authors have not performed a survival assay of flies with reduced MRP1 in Uro-cells? This could have answers if MRP1 is providing overall benefits to fly.

Minor points:

  1. Multidrug Resistance like Protein 1 has been mentioned in different versions, like Multidrug Resistance like Protein 1 and Multidrug-Resistance like Protein 1 with and without italics. Please choose and use one version.
  2. The first sentence of Abstract (Lines 9-12) is way too long. Please break it down into 2-3 sentences. 
  3. There is prevalent unnecessary usage of hyphen in various words like in-creased, detoxification, malpigh-ian etc. Please correct them.
  4. Please provide the BDSC stock number next to transgenic lines in the material and methods.
  5. Please mention the duration of the starvation assay being performed.
  6. Please mention the duration of FlyPad experiments, how flies were transferred into the arena? Iced or Co2 anesthesia? Please mention.
  7. Its mentions that data is plotted as Mean + SEM; however, Figure 1 is presented as box plots. Please correct.

Author Response

(The authors gave the same response as above.)

Reviewer 3 Report

In the manuscript "membranes-1223938", the authors examine the potential role of Drosophila MRP1, a member of the ABC transporter family capable to transport several organic anions across membranes, in regulating fly metabolism. Drosophila is a valuable model system to study this type of questions. It has less genetic redundancy than mammals and a rapidly increasing body of knowledge regarding the molecular, cellular and organismic mechanisms that control energy balance and metabolic homeostasis at a basic level.

Through a series of convincing and clear experiments, the authors show that MRP1 silencing in the Malpighian tubules, but not in other metabolically relevant tissues, induces global TAG accumulation in the fly and starvation resistance, two phenotypes that frequently correlate positively. This is accompanied by a reduction in feeding behaviors. Even though the manuscript does not define the mechanisms linking MRP1 activity in Malpighian tubules with a regulation of lipid metabolism, the discovery of this possible link is interesting. The authors also show that MRP1 silencing in Malpighian tubules leads to a decreased production of reactive oxygen species and increased oxidative resistance, an unexpected result for this type of proteins that is not clearly related to the metabolic phenotype.

As I have already mentioned, most of the experiments are well designed and the results convincing. However, I have a major experimental issue. All the results in the manuscript are based on the silencing of MRP1 in Malpighian tubules by interfering RNA and using a single RNAi line (P{TRiP.HMS01780}attP2). It is well known in the Drosophila field that a considerable proportion of RNAi lines display off-target effects. Thus, it is essential to show that the RNAi line used is really depleting MRP1 and that the results can be reproduced using other tools. In particular, there is another publicly available MRP1 RNAi line from VDRC (v105419), which has already been used to study MRP1 in another article ("Target Organ Specific Activity of Drosophila MRP ABCC1 Moderates Developmental Toxicity of Methylmercury" by Prince et al, 2014). Moreover, there are two hypomorphic mutations, also characterized in the mentioned paper, that can be helpful to corroborate the specificity of the described phenotypes. At a minimum, it should be shown that MRP1 silencing in Malpighian tubules using the alternate line v105419 also induces TAG accumulation in the fly.

Minor issue: in line 226, it shoukd read Figure 2E instead of 4E.

Author Response

(The authors gave the same response as above.)

Round 2

Reviewer 2 Report

I have no more comments. The authors have answered or edited their manuscript following my previous comments. 

Author Response

Dear Reviewer,

Thank you for your valuable comments. And to make sure our English is now correct we had a native English speaker read through and comment on spelling and grammar.

Reviewer 3 Report

The authors do not address my main concern regarding the manuscript membranes-1223938. They do not show that the single MRP1 RNAi line they used (P{TRiP.HMS01780}attP2) is really silencing MRP1 or that the described phenotypes are specific. All the conclusions in the manuscript are based on the phenotypes obtained after MRP1 silencing using P{TRiP.HMS01780}attP2 and thus, it is essential to demonstrate that this RNAi line is working as expected. As I mentioned in my previous report, there are several available tools that can be used to easily solve this issue.

Other observations:

In the new version, the authors have included an additional section in methods ("2.2. Successful knockdown of MRP") describing MRP1 silencing in the nervous system that is not used in Results. It is probably an error. 

Author Response

Dear Reviewer,

Thank you for your comments. Please see the attachment.

And to make sure our English is now correct we had a native English speaker read through and comment on spelling and grammar.

Round 3

Reviewer 3 Report

-

Author Response

Dear Reviewer,

Thank you. We had a native English speaker read through and comment on spelling and grammar. Please see the update version of Manuscript.